# Intraretinal Layer Segmentation Using Cascaded Compressed U-Nets

**DOI:** 10.3390/jimaging8050139

**Published:** 2022-05-17

**Authors:** Sunil Kumar Yadav, Rahele Kafieh, Hanna Gwendolyn Zimmermann, Josef Kauer-Bonin, Kouros Nouri-Mahdavi, Vahid Mohammadzadeh, Lynn Shi, Ella Maria Kadas, Friedemann Paul, Seyedamirhosein Motamedi, Alexander Ulrich Brandt

**Affiliations:** 1Experimental and Clinical Research Center, Max Delbrück Center for Molecular Medicine and Charité-Universitätsmedizin Berlin, Corporate Member of Freie Universität Berlin and Humboldt-Universität zu Berlin, 13125 Berlin, Germany; sunil-kumar.yadav@charite.de (S.K.Y.); rkafieh@gmail.com (R.K.); hanna.zimmermann@charite.de (H.G.Z.); josef.kauer@charite.de (J.K.-B.); friedemann.paul@charite.de (F.P.); seyedamirhosein.motamedi@charite.de (S.M.); 2Nocturne GmbH, 10119 Berlin, Germany; ella.kadas@nocturne.one; 3Glaucoma Division, Stein Eye Institute, David Geffen School of Medicine, University of California Los Angeles, Los Angeles, CA 90095, USA; nouri-mahdavi@jsei.ucla.edu (K.N.-M.); vmohammadzadeh@mednet.ucla.edu (V.M.); lshi@mednet.ucla.edu (L.S.); 4Department of Neurology, Charité-Universitätsmedizin Berlin, Corporate Member of Freie Universität Berlin and Humboldt-Universität zu Berlin, 10098 Berlin, Germany; 5Department of Neurology, University of California Irvine, Irvine, CA 92697, USA

**Keywords:** optical coherence tomography (OCT), intraretinal layer segmentation, retina, U-Net, deep learning

## Abstract

Reliable biomarkers quantifying neurodegeneration and neuroinflammation in central nervous system disorders such as Multiple Sclerosis, Alzheimer’s dementia or Parkinson’s disease are an unmet clinical need. Intraretinal layer thicknesses on macular optical coherence tomography (OCT) images are promising noninvasive biomarkers querying neuroretinal structures with near cellular resolution. However, changes are typically subtle, while tissue gradients can be weak, making intraretinal segmentation a challenging task. A robust and efficient method that requires no or minimal manual correction is an unmet need to foster reliable and reproducible research as well as clinical application. Here, we propose and validate a cascaded two-stage network for intraretinal layer segmentation, with both networks being compressed versions of U-Net (CCU-INSEG). The first network is responsible for retinal tissue segmentation from OCT B-scans. The second network segments eight intraretinal layers with high fidelity. At the post-processing stage, we introduce Laplacian-based outlier detection with layer surface hole filling by adaptive non-linear interpolation. Additionally, we propose a weighted version of focal loss to minimize the foreground–background pixel imbalance in the training data. We train our method using 17,458 B-scans from patients with autoimmune optic neuropathies, i.e., multiple sclerosis, and healthy controls. Voxel-wise comparison against manual segmentation produces a mean absolute error of 2.3 μm, outperforming current state-of-the-art methods on the same data set. Voxel-wise comparison against external glaucoma data leads to a mean absolute error of 2.6 μm when using the same gold standard segmentation approach, and 3.7 μm mean absolute error in an externally segmented data set. In scans from patients with severe optic atrophy, 3.5% of B-scan segmentation results were rejected by an experienced grader, whereas this was the case in 41.4% of B-scans segmented with a graph-based reference method. The validation results suggest that the proposed method can robustly segment macular scans from eyes with even severe neuroretinal changes.

## 1. Introduction

Optical coherence tomography (OCT) is a noninvasive in vivo imaging modality, which is able to acquire 3D volume scans of the retina with micrometer resolution [1]. The retina is part of the central nervous system, and OCT-derived intraretinal layer thickness or volume measurements, which are typically measured within the macular region, are promising noninvasive biomarkers querying neuronal structures. For instance, the combined macular ganglion cell and inner plexiform layer (GCIPL) shows thinning in eyes affected by optic neuritis (ON) [2,3], multiple sclerosis (MS) [4], neuromyelitis optica spectrum disorders (NMOSD) [5] and Alzheimer’s disease [6]. GCIPL is a potential biomarker predicting disease activity in patients with early MS [7,8] and of cognitive decline in Parkinson’s disease [9]. The macular inner nuclear layer (INL) may be affected by inflammatory activity in MS and related disorders [10] or Müller cell damage in NMOSD [5,11,12]. In glaucoma, thinning of inner retinal layers such as GCIPL and the retinal nerve fiber layer (RNFL) may appear before visual function loss and is a potential biomarker for early diagnosis [13]. Notably, these applications rely on changes in layer thickness, while the overall configuration of the retina is mostly unaltered, which separates their segmentation needs from disorders with clear microscopic retinal pathologies such as age-related macular degeneration (AMD) or diabetic retinopathy (DR) [14]. Indeed, retinal scans with comorbid eye diseases such as AMD and DR are typically excluded from their use as biomarkers in neuronal diseases such as MS [15].

Intraretinal layer segmentation is not a trivial task as layer changes can be small, while texture gradients can be weak and influenced by noise [16]. While many OCT manufacturers provide device-integrated segmentation of some intraretinal layers, results are often noisy and regularly require manual correction, which in itself is challenging for clinical use and difficult to handle in research settings when multiple graders review macular OCT images [16,17]. A reliable and robust algorithm for intraretinal layer segmentation that requires no or minimal manual correction is an unmet need to foster reliable and reproducible research as well as clinical application [18].

In this paper, we propose a cascade of compressed convolutional neural networks with a U-Net-like architecture for fully automated intraretinal layer segmentation (CCU-INSEG). We compress the U-Net models to reduce required resources without losing a significant amount of accuracy in the segmentation process. The cascaded architecture helps the algorithm to increase its robustness with regard to different types of OCT images without adding too many parameters in the overall network. We validate the method using internal and external data on a voxel, measurement parameter and scan level.

### State of the Art

Intraretinal layer segmentation of OCT-derived macular volume scans can be divided into two categories, mathematical modeling-based methods and machine learning (ML)-based methods, with the latter recently extending towards deep learning.

Mathematical modeling-based methods follow a set of rules based on well-defined deterministic mathematical and image analysis concepts to segment intraretinal layer boundaries. These methods have been the predominant approach in the previous decade and are not within the scope of this paper.

Data-driven or ML-based methods learn features from image data sets automatically and have produced reliable and accurate measurements in several retinal imaging applications [14]. In the first paper proposing deep learning-derived methods for intraretinal layer segmentation, Fang et al. [19] used a Cifar-style convolutional neural network (CNN) to create class labels and probability maps, followed by a graph theory and dynamic programming step to derive final layer boundaries in OCT images from patients with age-related macular degeneration (AMD). Shortly thereafter, Roy et al. [20] proposed an end-to-end fully convolutional framework to perform intraretinal segmentation, which was also able to delineate fluid collection in diabetic retinopathy. The authors avoided all deterministic pre- and postprocessing and performed the entire segmentation procedure using a fully convolutional network inspired by DeconvNet [21] and U-Net [22] with an encoder–decoder architecture, skip connections and unpooling layers. Kugelman et al. [23] used recurrent neural networks (RNN) instead of CNNs for 2D image segmentation by formulating image patches as a sequential problem. Unlike CNNs, RNNs use interim output as input in the next iteration of computations, thereby utilizing a memory network in a specific classification task on sequential data [24].

Several recent publications make use of the U-Net design, which was introduced to target problems in medical image analysis [22]. U-Net’s symmetrical architecture consists of a contracting path to capture context and an expanding path that enables precise localization, linked by skip connections. Mishra et al. [25] proposed a method similar to Fang et al. but relying on U-Net for the generation of probability maps and shortest path segmentation for final boundary detection. He et al. [26] proposed an intraretinal segmentation method using two cascading U-Nets, the first one preforming the actual intraretinal segmentation task, and the second one then reconstructing 3D surfaces. Combining segmentation with surface reconstruction using CNN was initially suggested by Shah et al. [27] and has the benefit that incorrect segmentation results in 2D planes are corrected when reconstructing a continuous surface. This approach has been recently combined into a single unified framework by the same group [28]. Pekala et al. [29] proposed a method based on DenseNet, which makes use of more extensive skip connections but is otherwise similar in design to U-Net for boundary pixel segmentation, followed by a Gaussian process regression for surface extraction. Another extension of the U-Net concept for retinal segmentation was proposed by Liu et al. [30], who used a nested U-Net shape for multi-scale input and output; this approach performed favorably based on accuracy in comparison to traditional U-Net architectures. Instead of U-Net, Li et al. [31] proposed a method utilizing a combination of the Xception65 network and an atrous spatial pyramid pooling module for retinal boundary detection. Cascaded network architectures provide better generalization and are therefore suitable for further improvements in robustness and efficiency [32]. In the literature, cascaded networks are successfully applied to solve several problems—for example, time-series prediction [33], human–object interaction modeling [34] and medical image segmentation [35]. In this paper, we have also utilized the cascaded network framework with compressed U-Net architecture.

The segmentation accuracy of these previously proposed deep learning-based methods is favorable, with more recent and complex approaches performing better than early methods utilizing simpler network architectures. Hamwood et al. [36] showed that network design choices such as patch size may have a strong influence over network performance in retinal segmentation tasks. In contrast, in real-world medical applications, the selection of training data, meaningful pre- and postprocessing utilizing domain knowledge and computational costs are highly relevant. The latter favors simpler network architectures, because complex architectures with multiple parameters are prohibitive for environments with limited resources. Methods implementing lean networks and specific postprocessing for segmentation errors, e.g., by reconstructing surfaces, can achieve high segmentation efficiency while being performant [28].

One way in which performance optimization may be achieved is by compressing network architectures—for example, by architecture pruning, weight quantization or knowledge distillation [37]. The latter was recently used by Borkovkina et al. [38] to propose a retinal segmentation method, which utilizes a performance-optimized smaller network that was trained on a complex original segmentation network as teacher network.

We here utilize the compression of a U-Net-style network based on architecture pruning, which leads to substantial performance improvements while not suffering in accuracy. We further utilize appropriate pre- and postprocessing to reduce errors and achieve high segmentation accuracy, by introducing an adaptive hole filling process. Of note, our method was also trained on more diverse data than previous approaches, improving its readiness for immediate application rather than merely being a methodological proposal. Together, these contributions highlight important pathways from research-driven proof-of-concept implementations towards application.

## 2. Materials and Methods

The proposed method starts with preprocessing an input volume scan to remove undesired components from each B-scan (Figure 1). It then performs intraretinal segmentation using a cascaded network, where each stage is a compressed U-Net-like network. Finally, during postprocessing, we derive continuous surfaces and introduce anatomical constraints. The term cascaded network refers to the combination of two networks in a sequential manner and each network is a compressed U-Net-like architecture, which is a lighter version of U-Net, to minimize the computational resources. In the following, we explain each component of the pipeline in detail.

### 2.1. Segmentation Method

#### 2.1.1. Preprocessing

Let us consider an input OCT volume scan sampled from the macula consisting of several B-scans. During data acquisition, artefacts are inevitable due to various internal and external factors. These artefacts may include noise, which is responsible for a low signal-to-noise (SNR) ratio, cut scans (upper and lower cuts) and improper focus [15]. Additionally, parts of the optic nerve head (ONH) frequently appear in macular scans, as shown in Figure 1 (the input image). These artefacts are not desirable and affect the performance of segmentation algorithms.

To automatically detect and remove these artefacts, we employ a deep learning-based quality control method described and validated in detail previously [39,40]. Briefly, the automatic quality analysis (AQuA) method combines the detection of center, signal quality and image completeness artifacts. These features are evaluated before segmentation, and only if the input volume is free from cuts, focused around the macular center, and have acceptable signal quality, the segmentation operation will be performed. AQuA further allows some correction of quality issues: the scan center is corrected using parabola fitting in the foveal region. Furthermore, based on the computed foveal center, B-scans are cropped in both A-scan and B-scan directions to cover only a 6x6mm region around the center for both training and prediction [40].

#### 2.1.2. Cascaded and Compressed U-Net Networks

Our segmentation method consists of two cascaded U-Net-like architectures, as shown in Figure 1. The first network, termed RS-Net (retinal segmentation network), is responsible for segmentation of the retina from a full OCT B-scan. In its simplest form, this is the retinal tissue between the inner limiting membrane (ILM) and Bruch’s membrane (BM) as boundaries. The trained network takes a preprocessed OCT B-scan as input and produces a three-class output: purple region (above ILM), green region (retinal region) and yellow region (below BM) (Figure 1). The output of RS-Net and the original input serve as the input for the second network. The second network, the intraretinal layer segmentation network (IS-Net), takes the two-channel input and generates a 9-class output providing boundaries for 8 different intraretinal layers and membranes: ILM—inner limiting membrane, mRNFL—macular retinal nerve fiber layer, GCL—ganglion cell layer, IPL—inner plexiform layer, GCIPL—macular ganglion cell/inner plexiform layer, INL—inner nuclear layer, OPL—outer plexiform layer, ONL—outer nuclear layer, ELM—external limiting membrane, MZ—myoid zone, EZ—ellipsoid zone, OS—outer segment, RPE—retinal pigment epithelium, OL—outer layer and BM—Bruch’s membrane (Figure 2).

The original U-Net is a fully CNN that consists of a contracting path (encoder) and an expansive path (decoder) [22]. Paths are connected at multiple places by skip connections, a key aspect of U-Net that allows the network to restore spatial information that was lost during the pooling operation. In general, U-Net is capable of producing high-fidelity segmentation results and most of the state-of-the-art segmentation methods are based on U-Net architecture and focused on high accuracy (see above). However, the deep architecture of U-Net requires considerable computational resources, which restricts the deployment of the model on systems with limited processing power. To find a desirable balance between processing needs and accuracy, we studied compressed variations of this network with regard to channel depth and filter size.

Figure 3 shows the architecture of the proposed compressed U-Net. Similar to U-Net, the proposed method has contracting and expansive paths. The contracting or encoder part has four stages and, at each consecutive stage, the channel depth is doubled and the input size is halved (via max pooling). Additionally, dropout layers are added after the third and the fourth stages to avoid overfitting. In the decoder part, the inverse of the encoder part occurs, except that it has two additional blocks: merging and output convolution layers with a kernel size of 1, as shown in Figure 3. The terms *h* and *w* represent the size of the preprocessed image. The number of the input channels is represented by *n* and it is clear from Figure 1 that n=1 for RS-Net and n=2 for IS-Net. Optimal architecture parameter values (*h*, *w*, *f*, *c*) are explained in Section 2.3.

#### 2.1.3. Loss Function

For end-to-end training of the proposed network, a suitable loss function will be the key to a high-fidelity segmentation outcome. In the case of OCT layer segmentation, one of the main challenges would be to tackle the background–foreground class imbalance. As shown in Table 1, 7 out of 9 classes are highly imbalanced and having less than 10% of foreground pixels with respect to all pixels in the training data. These imbalance factors should be corrected to avoid false negatives, as shown in Figure 4b (red rectangular region). Figure 4b shows the outcome from the combination of Tversky [41] and CCE (categorical cross-entropy) loss functions.

Focal loss is a robust loss function to provide a better segmentation outcome with extremely foreground–background imbalanced training data. It is a dynamically scaled version of cross-entropy to down-weight the background pixels and focus the model on foreground pixels (layers). Focal loss is defined as [42]:(1)$$E_{fl}=\alpha_{t}{\left(1-p_{t}\left(x\right)\right)}^{\gamma }log\left(p_{t}\left(x\right)\right),$$
where $p_{t}$ is defined as:$$p_{t}\left(x\right)=\left\{\begin{array}{cc} p_{c}\left(x\right), & \mathrm{if}\;g_{c}\left(x\right)=1\\ 1-p_{c}\left(x\right), & \mathrm{otherwise}, \end{array}\right.$$
where *x* is the pixel value from the retinal region Ω. The terms $g_{c}\left(x\right)$ and $p_{c}\left(x\right)$ show the ground truth and the predicted probability at *x* from the class *c*, respectively. Furthermore, $\alpha_{t}$ and γ are modulating factors and, in our experiment, we used default values of α=0.25 and γ=2. As can be seen from Figure 4c, the outcome improves compared to Tversky loss [41]. However, we can see a significant amount of false negatives. To improve the focal loss outcome, we replace $\alpha_{t}$ with $w_{t}$. The weighting fact $w_{t}$ and modified focal loss are defined as:(2)$$w_{t}=1-\frac{n_{f}}{n_{t}},\;E_{wfl}=w_{t}{\left(1-p_{t}\left(x\right)\right)}^{\gamma }log\left(p_{t}\left(x\right)\right),$$
where $n_{f}$ and $n_{t}$ represent the number of pixels in the foreground and in the image, respectively. For our training, we have used weighted focal loss $E_{wfl}$, where each class is weighted based on the corresponding foreground–background pixel ratio. The improvement in the outcome is visualized in Figure 4.

#### 2.1.4. Postprocessing

At the last stage of the pipeline, we apply postprocessing on the segmentation outcome to remove artefacts. First, adaptive hole filling removes small holes from the background and the foreground of the segmentation outcome as shown in Figure 5. These holes occasionally appear because of the misclassification of individual or a few pixels during the segmentation process. To remove small holes, we use morphological operators with an adaptive threshold. The value of the threshold is computed differently for each class of the network’s outcome. RS-Net generates a three-class output, and for upper and lower classes (above ILM and below BM), the threshold value is $n_{n}/f$, where $n_{n}$ is the number of non-zero pixels in this class image and *f* is a constant factor with a default value of 2. Additionally, the value of *f* will keep increasing by 1 until $n_{n}/f<n_{z}$, where $n_{z}$ is the number of zero pixels. The same process is repeated for the upper and lower class of the IS-Net outcome. For the internal layer classes of the networks, the default value of *f* is 10 and it is increased by 1 until $n_{n}/f<n_{z}$.

After hole filling, different layers from the multi-channel outcome of IS-Net are extracted. To detect the boundary in each class image, we search the first non-zero entry in each column. After layer extraction, we remove outliers from the layer surfaces based on the Laplacian of the thickness values. Furthermore, we recompute missing values using Piecewise Cubic Hermite Interpolating Polynomial (PCHIP) interpolation. Lastly, an isotropic smoothing with 3×3 kernel size is applied to each layer surface to remove small ripples and to generate smoother boundaries.

#### 2.1.5. Implementation

The method was fully implemented in Python. Network training was performed with Python (v3.6) using Keras (v2.3) and TensorFlow (v2.2) libraries on a Linux server with two Intel Xeon Gold 6144 CPUs (Intel Corporation, Santa Clara, CA, USA) and two NVIDIA GeForce GTX1080 Ti GPUs (NVIDIA Corporation, Santa Clara, CA, USA).

### 2.2. Data

To train the proposed method, we selected macular OCT volume scans that were obtained with Spectralis spectral-domain OCT (Heidelberg Engineering, Heidelberg, Germany). All data were selected from the institute’s OCT image database at Charité—Universitätsmedizin Berlin (CUB).

In our previous experience with intraretinal layer segmentation, we noticed that macular OCT volume scans with low, yet acceptable, signal to noise ratios, as well as scans from patients with very thin RNFL, needed extensive manual segmentation correction. Hence, from a pool of macular scans from healthy controls and patients with clinically isolated syndrome (CIS), MS and NMOSD, we selected a total of 445 OCT volume scans of 254 eyes, the majority of which had a low to average signal to noise ratio, thin RNFL and microcystic macular pathology, an occasional OCT finding in eyes from patients with various forms of optic neuropathy such as autoimmune optic neuropathies [43]. All selected volumes were centered on the fovea, covering $25^{\circ }\times 30^{\circ }$ with different resolutions (61/25 B-scans and 768/1024/1536 A-scans per B-scan).

All scans underwent automatic quality control to avoid data with insufficient quality for training and validation purposes [39,40]. In the end, we selected 17,458 B-scans for training, 3081 for validation and 208 for testing.

#### 2.2.1. Manual Correction

For the manual correction of segmented boundaries, we used standardized procedures developed during an international study on intraretinal segmentation with several reference centers [17]. Data were manually segmented by multiple graders, who were all trained and experienced in applying these procedures. All grading was supervised by experienced graders (HGZ, EMK). Intraretinal segmentation was performed using the SAMIRIX toolbox [44], which uses OCTLayerSegmentation from AUtomated Retinal Analysis Tools (AURA) developed by [45], and then manually corrected by experts using the same toolbox. A detailed description of this process, including validation, is outlined in [44].

#### 2.2.2. Validation Data

To validate the method in a multicenter format, additional unseen macular volume scans from our center (CUB), as well as two other centers—the University of California, Los Angeles (UCLA) and Johns Hopkins University (JHU)—were added.

**CUB:** Twenty-five additional unseen macular 3D OCT scans of 24 eyes were included in this study from our center (CUB). The volumes were from HCs and patients with MS and NMOSD, covering $25^{\circ }\times 30^{\circ }$ macular area using 61 B-scans and 768 A-scans per B-scan. All volumes underwent segmentation and manual correction using the SAMIRIX toolbox [44]. Two scans were rejected by our quality check because of the low signal to noise ratio.

**UCLA:** In order to ensure the quality of the developed method on OCT images from patients with other optic neuropathies, which were not part of the training data, we included a second data set for testing from the University of California, Los Angeles (UCLA). The data set consists of 12 OCT volume scans of 12 eyes from glaucoma patients or healthy subjects, each with 61 horizontal B-scans and 768 A-scans per B-scan, covering $25^{\circ }\times 30^{\circ }$ centered on the fovea. The OCT volumes in this data set were segmented and then manually corrected using the SAMIRIX toolbox for comparison against the developed method. The volume scans in this data set were cropped to 5.5×5.5 mm square in the preprocessing step to ensure complete exclusion of ONH from macular scans.

**JHU:** To validate our method further, we applied the developed segmentation method on a publicly available OCT data set from Johns Hopkins University (JHU) [26]. Fifteen volume scans of 15 eyes were selected from this data set, which are a mix of HC and MS data. These scans were segmented and manually corrected by its publisher, so we compared our segmentation results against their manual delineation. The volume scans in the JHU data set cover $20^{\circ }\times 20^{\circ }$ of the macula using 49 B-scans and 1024 A-scans per B-scan.

### 2.3. Optimization

To obtain optimal values of filter kernel size *f* and channel depth *c*, we performed compression with different values of *f* and *c*, as shown in Table 2. The network’s channel depth should be proportional to the complexity of task. From Figure 1, it can be seen that RS-Net produces a three-channel output (o=3), which is a less complicated task compared to generating a 9-class output (IS-net).

For RS-Net, we start with two different values of the depth channel (c={4,8}), which increases up to four times the input size, as shown in Figure 3. RS-Net is responsible for retinal segmentation, and it should not focus on minor details. Therefore, we set the kernel size to 5×5 (f=5), which helps the network to focus on significant features. To check the performance of the RS-Net with different values of *c*, we use a smaller data set (1000 B-scans) for training, validation and testing along with the loss function, which is explained in Section 2.1.3. As can be seen from the Table 2, the performance of RS-Net is similar in terms of the dice similarity coefficient (DSC) for both values of *c*. However, the number of model parameters is almost four times less with c=4 in comparison to c=8. Therefore, f=5 and c=4 are the optimal configuration for RS-Net with respect to model size and performance metrics.

In contrast to RS-Net, IS-Net is responsible for separating 8 layers (9 classes) with the need to detect small details. Different configurations of IS-Net along with the performance metrics are shown in Table 2. The two combinations of channel depth and kernel size (f,c={5,16}&f,c={3,32}) produce the best performance metrics. However, the first configuration (f=5,c=16) has a smaller number of model parameters and, therefore, is the preferable configuration of IS-Net for our purpose.

In contrast to U-Net, the proposed architecture has fewer stages. U-Net consists of 5 stages ranging from 64 to 1024 channel depths, and our networks consist of only 4 stages with variable channel depths. Table 3 shows the direct comparison with U-Net and deep residual U-Net ([46]) in terms of the number of model parameters, DSC and the prediction time. ResU-Net performs slightly better compared to the original U-Net and our proposed method. It is worth mentioning that the above comparison is performed at the architectural level. If U-Net is used in the cascaded setting as in [47], then the computational complexity will be high. Although ResU-Net has not been used in the cascaded setting so far in the state-of-the-art methods, it has better potential because of the low number of parameters and slightly better performance compared to the original U-Net. However, it still has more parameters than the proposed architecture.

As a result, RS-Net is almost 392-times smaller compared to the original U-Net in terms of parameters and model size. Similarly, IS-Net is almost 26-times smaller compared to U-Net. The comparison of prediction time could be challenging because it depends on several factors, including hardware, programming style, libraries, etc. However, it is clear that the smaller network will need smaller computational resources and will be faster compared to U-Net from the running time perspective. Table 3 shows the training time for 1 epoch and the mentioned time is taken on a Linux server with two Intel Xeon Gold 6144 CPUs and two NVIDIA GeForce GTX1080 TI graphical processing units (GPUs). Furthermore, the prediction time is recorded only on a Core-i7 CPU. From Table 3, it is clear that the proposed networks maintain the balance between the need for computational resources (lower running time or both training and prediction) and accuracy. Our method produces a quite similar DSC score to ResU-Net and the original U-Net. Furthermore, the number of model parameters indicates the memory requirement during the training and inference.

Compared to two-stage cascaded state-of-the-art methods, our CCU network is much smaller regarding parameters and model size, with almost similar or better accuracy. For example, Ref. [35] proposed a cascade of two U-Net-like architectures, where each network is a combination of the original U-Net with additional layers. Therefore, the number of parameters in each network is more than 31M (for the input image size of 512×512), which is much larger compared to our CCU network. Similarly, [26] used a modified U-Net for the first network with a double channel depth compared to our CCU network.

### 2.4. Statistical Analysis

Intraclass correlation coefficient (ICC) and lower and upper confidence intervals were calculated based on the variance components of a one-way ANOVA, using the ICC package of R. Mean absolute error (MAE), standard deviation and DSC are computed using Python.

## 3. Results

### 3.1. Training Analysis

End-to-end training was performed using 17,458, 3081 and 208 B-scans for training, validation and testing, respectively. To minimize the loss function (Equation (Equation 1)), the Adam optimizer was used with a fixed learning rate of 0.001. We set the number of iterations to 80 for both networks with early stopping criteria. For early stopping, we monitored the validation accuracy.

As can be seen from Table 4, the optimal and most stable accuracies are reached in 23 and 11 epochs for RS-Net and IS-Net, respectively. Additionally, Table 4 consists of different training- and testing-related metrics of the proposed CCU network. Furthermore, Figure 6 represents a visual outcome with the data from three different centres.

### 3.2. Comparison against Device-Implemented Segmentation

We then compared the proposed method and standard device segmentation without further correction. Table 5 shows MAE (SD) between manually corrected gold standard segmentation (15 volume scans of 15 eyes) and the device segmentation (first row). Similarly, the second row shows MAEs (SD) between the proposed method’s outcome and gold standard segmentation. Our method has almost 2.5 times lower MAEs and SDs compared to the device segmentation.

### 3.3. Voxel-Wise Comparison against Multicenter Data

We also validated our method with multi-center data, which included data from Charité—Universitätsmedizin Berlin (CUB), the University of California, Los Angeles (UCLA) and publicly available data from Johns Hopkins University (JHU).

DSC values were computed between manually corrected segmentation using the SAMIRIX toolbox and the proposed method’s outcome for nine classes for CUB and UCLA. The corresponding values are reported in Table 6, where vitreous is the region above ILM and b-BM is the region below BM, as shown in Figure 7. As can be seen from Table 6, the first and the last classes (vitreous, b-BM) yield the highest DSC values (close to 1) as the differences between the numbers of background and foreground pixels are small. Moreover, OPL has the lowest value of DSC since this is one of the thinnest layers and the difference between the numbers of background and foreground pixels can be large. Compared to the state-of-the-art methods, our method produces better overall DSC scores (CUB-0.95, UCLA-0.94 and JHU-0.92), except [38,48], where validations are performed using HC data only.

Table 7 shows layer-based MAE computed between CCU-INSEG’s outcome and the manually corrected gold standard segmentation for the CUB and UCLA data sets and the manually corrected segmentation by the publishers for the JHU data set. As can be seen, the proposed method produces a high-fidelity segmentation with regard to the gold standard segmentation. Errors for different layers are in sub-pixels, except for the INL-OPL boundary in the JHU data.

We would like to emphasize an important point here that our method is trained only on data from one center (CUB). The CCU network was not trained on the data either from UCLA or from JHU. However, the performance of our method is quite similar for all these data in terms of MAE (sub-pixel level). Figure 6 shows the effectiveness of the proposed method against different challenges from these three different data sets. Figure 6 (top) is a noisy scan (CUB data), Figure 6 (middle) is a scan with microcysts from JHU data, and UCLA data have a quite thin RNFL on one side of the macula. As can be seen from Table 7, MAE values for BM and INL-OPL are on the higher side (close to 1 pixel) in the JHU data, where manual segmentation was not performed by the same expert as in CUB and UCLA, and it is quite possible that the segmentations from different experts are not exactly the same [17].

### 3.4. Reliability and Fidelity of Derived Thickness and Volume Parameters

In this section, we compute the standard OCT parameters based on the segmented layers using the proposed method. Here, we are focusing on five standard thickness parameters within a 5 mm circular region from the foveal center. The parameters are described in Figure 2, and additionally we provide data for the ganglion cell complex (GCC, tissue between ILM and IPL-INL boundary) and total macular thickness (TM, between ILM and BM).

The test–retest reliability of the proposed method is evaluated on a data set that has three repeated measurements of 30 healthy eyes. The intra-class correlation coefficient (ICC) for the standard parameters varied from 0.93 (mRNFL) to 0.99 (GCIPL, GCC and total macula) for both thicknesses and volumes, as shown in Table 8. As shown in Table 8, mRNFL has the lowest ICC because mRNFL is one of the thinnest layers in the retina. Therefore, the measurement of this layer can be easily contaminated by noise components, which leads to the lower value of ICC.

To check the fidelity of our method, we use the same multi-center data sets with the gold standard segmentation. Furthermore, the standard parameters are computed on both segmentations (the gold standard and the outcome of the proposed method). Table 8 (the last three rows) shows MAEs computed between parameter values based on the gold standard segmentation and our outcome. For CUB and UCLA data, the errors are limited up to 2 μm for thickness parameters. For JHU data, INL-related parameters are not quite similar. It is possible that the higher error is because of the different manual corrections at different centers.

### 3.5. Comparison against State-of-the-Art Methods

For comparison against state-of-the-art methods, we randomly selected 20 macula volume scans of 20 eyes from patients with NMOSD from a recent publication [44]. Eyes from NMOSD patients regularly show severe neuroaxonal retinal damage, resulting in problematic image quality and/or larger neuroaxonal changes. While this does not reflect performance in a random data set, it does test the segmentation accuracy in difficult data. For comparison, we chose OCTLayerSegmentation version 2.11 from AURA (random forest-based boundary classification, [45]), OCTExplorer version 3.8.0 (OCTExp) from The Iowa Reference Algorithms (Retinal Image Analysis Lab, Iowa Institute for Biomedical Imaging, Iowa City, IA, USA) (graph-cut-based segmentation, [49,50,51]) and the PyTorch implementation of RelayNet (deep learning-based segmentation, [20]), three of the most highly cited segmentation methods, which are also made publicly available by their authors.

Table 9 shows that our method outperforms these methods in this problematic data set. AURA performs weakly for mRNFL and GCIPL, which has been reported before [26]. OCTExp shows weak agreement with the reference segmentation in all investigated layers, likely caused by over-smoothing and disagreement in several boundary positions. RelayNet produced several false positive and false negative pixels in the prediction, leading to overall low DSC values. In Figure 8, we show a sample image, exemplifying the segmentation performance on an exceptionally noisy B-scan. To explore segmentation performance on scans with retinal pathologies, we tested all methods on sample scans with geographic atrophy, Susac syndrome and Drusen (Supplementary Data). All methods failed to produce acceptable results in these scans.

### 3.6. Manual Correction Timing Comparison

Finally, we compare the correction time required to correct the segmentation obtained from the proposed method and the state-of-the-art methods. An experienced grader quality-controlled the segmentation outcome of 15 volume scans from four different methods: HeyEx (device segmentation), OCTExp [49], AURA [45] and CCU (the proposed method). The same grader performed segmentation correction if needed and recorded the corresponding time needed for manual correction (Table 10). Additionally, MAE (mean absolute error) is computed between the corrected and uncorrected segmentation. As can be seen from Table 10, the proposed method requires the minimum average correction time and MAE is also much smaller compared to the state-of-the-art methods. Moreover, only 2 volumes out of 15 required any correction from our method. On the other hand, the other methods required correction for at least 40% of data.

## 4. Discussion

In this paper, we propose a method for the automatic intraretinal layer segmentation of macular OCT images (CCU-INSEG). The focus of this work was to develop an efficient and high-fidelity segmentation algorithm, which requires little to no manual correction and can be easily deployed on devices with limited memory and computational power. The proposed method is a two-stage segmentation algorithm, where the retinal region is separated in the first stage and intraretinal layer segmentation is performed in the second stage. We further purposefully optimized the method to detect changes in OCT from neurological disorders, which typically show layer thickness differences, whereas scans with macroscopic retinal findings from primary eye disorders are of little relevance in this application.

In a direct comparison, our method outperforms several state-of-the-art methods in low-quality scans, substantially reducing the need for manual correction of segmentation results, which was the main goal of our study. This is likely a result of the optimized training set rather than the network architecture. Our study further demonstrates that good segmentation accuracy can be achieved using compressed networks, but meaningful domain-specific processing and training data selection. We selected a heterogeneous training data set, which included healthy controls and different autoimmune inflammatory and neurodegenerative diseases of the central nervous system. Thorough quality control was performed on each image, and we selected images representing a range of real-world quality, including low-quality images. As such, our method is specialized towards retinas with thickness changes from neuroinflammation and neurodegeneration, but without structural retinal changes from primary eye diseases.

We performed voxel-wise evaluation of our proposed method on multi-center data, which included data from three different centers (CUB, UCLA and JHU). Here, our method produces almost equally good segmentation (sub-pixel level MAE) as the reference segmented data from these centers. Importantly, one data set was from patients with glaucoma, and this disease was not part of the training data set. Nonetheless, performance was comparable, confirming that our method can extend beyond its training domain. On the other hand, while generally good, noise was higher when validating against JHU data, mainly for the INL-OPL boundary (above a pixel) and BM (close to a pixel). We hypothesize that this was caused by different graders manually correcting each data set. Previously, we have shown that manual grader segmentation can systematically differ within and between centers, which is one of the core motivations for a fully automatic, reproducible segmentation method [17].

While the outcomes of our proposed method are encouraging for applications in neurology and neuro-ophthalmology, ours and other proposed methods are still limited in their generalizability, making each method only robust in a select scenario intended by its investigators. Here, we aimed for excellent performance in scans from patients with autoimmune optic neuropathies such as MS, NMOSD and MOGAD, whereas eyes with primary retinopathies and eye disorders are typically excluded in this scenario [15]. However, generalization towards primary eye disorders presumably constitutes a problem of training data and less the underlying architecture. Consequently, improving the representation of primary eye diseases within the training data, including class expansions, is a promising route to further generalizability and clinical application.

## Figures and Tables

**Figure 1 jimaging-08-00139-f001:**
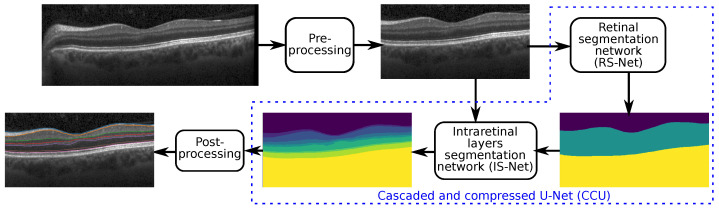
The pipeline of the proposed method consists of four steps: preprocessing, retinal region segmentation, intraretinal layer segmentation, and postprocessing.

**Figure 2 jimaging-08-00139-f002:**
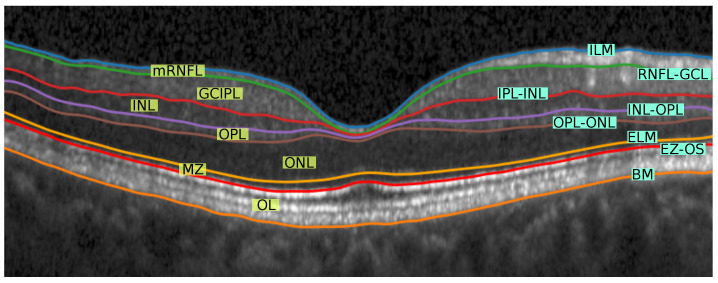
Visual representation of different retinal layers and boundaries after segmentation. Our method segments 9 different regions (7 highlighted with yellow background on the **left** side of the image and other two region are: above ILM and below BM) from 8 different retinal boundaries (highlighted with cyan background on the **right** side of the image). Layer abbreviations are positioned within their respective layer; boundary abbreviations are positioned on top of their respective segmentation lines. Abbreviations: ILM—inner limiting membrane, mRNFL—macular retinal nerve fibre layer, GCL—ganglion cell layer, IPL—inner plexiform layer, GCIPL—macular ganglion cell/inner plexiform layer, INL—inner nuclear layer, OPL—outer plexiform layer, ONL—outer nuclear layer, ELM—external limiting membrane, MZ—myoid zone, EZ—ellipsoid zone, OS—outer segment, RPE—retinal pigment epithelium, OL—outer layer and BM—Bruch’s membrane.

**Figure 3 jimaging-08-00139-f003:**
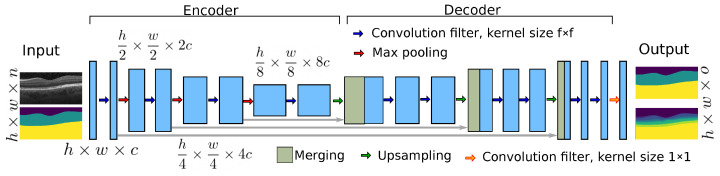
The general architecture of the proposed compressed U-Net. RS-net and IS-Net both have the same architecture with different channel depth *c*, kernel size *f*, number of input channels *n* and number of output classes *o*.

**Figure 4 jimaging-08-00139-f004:**
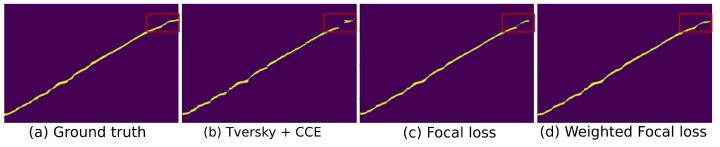
The comparison of different loss functions on a single class (INL). As can be seen, the weighted focal loss is capable of producing the minimum false positives.

**Figure 5 jimaging-08-00139-f005:**
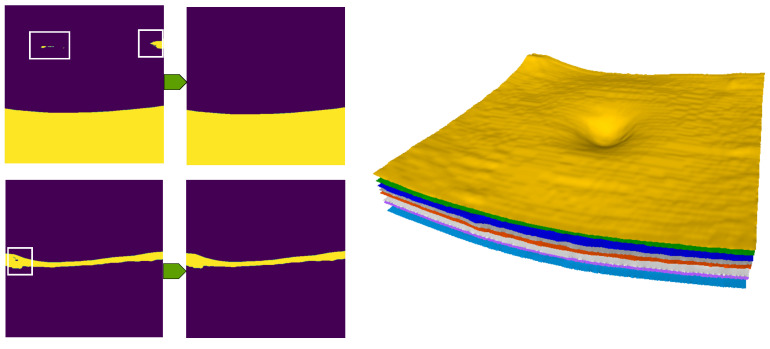
The first column shows small holes, which appear during the classification in both networks. The second column consists of images without holes. The last column shows 3D reconstructed surfaces of different layers.

**Figure 6 jimaging-08-00139-f006:**
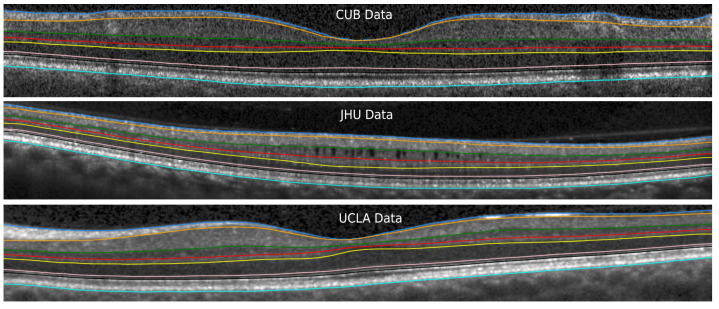
The visualization of the segmentation lines computed by the proposed method for three different data sets.

**Figure 7 jimaging-08-00139-f007:**
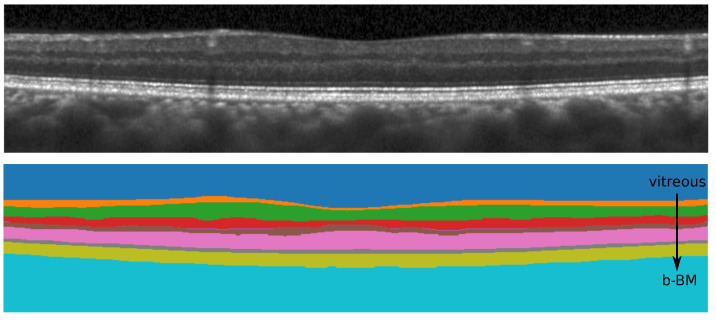
All nine different classes segmented by the proposed method.

**Figure 8 jimaging-08-00139-f008:**
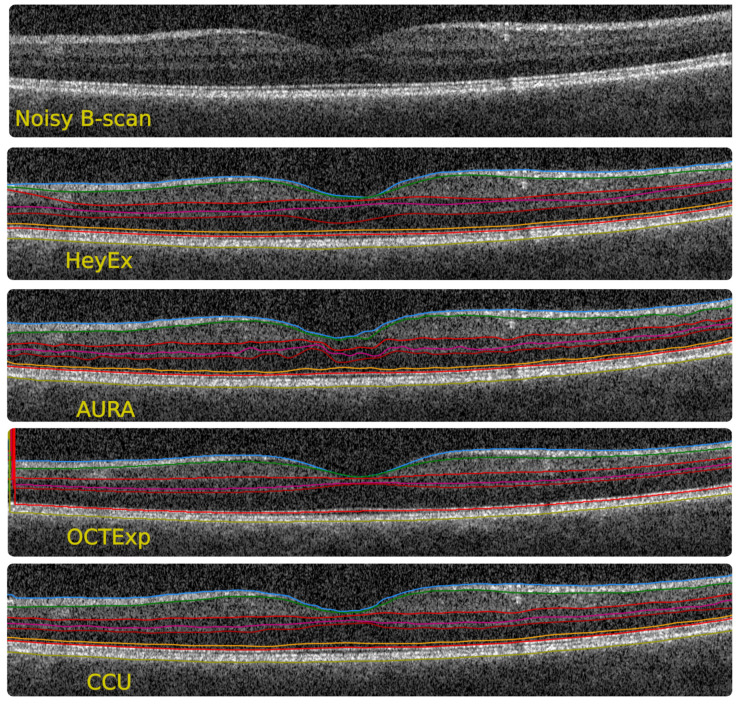
Example segmentation with a low-quality scan to show the performance of the proposed method compared to state-of-the-art methods (HeyEx—device segmentation, AURA [45], OCTExp [49]).

**Table 1 jimaging-08-00139-t001:** Background and foreground pixel imbalance: the table shows the percentage of foreground pixels (FP) with respect to the total pixels in the training data for each class.

Data	Vitreous	mRNFL	GCIPL	INL	OPL	ONL	MZ	OL	b-BM
FP	33%	5%	9%	5%	3%	9%	3%	8%	24%

**Table 2 jimaging-08-00139-t002:** The compression experiments with different architectural parameters for both RS and IS networks. The selected configuration for each network is highlighted in bold.

Network Architecture	*f*	*c*	Validation DSC	Test DSC	# Parameters	*o*	*n*	*h*	*w*
RS-Net	**5**	**4**	**0.9979**	**0.9976**	**79K**	**3**	**1**	**512**	**512**
	5	8	0.9984	0.9975	0.31M	3	1	512	512
	**5**	**16**	**0.9933**	**0.9933**	**1.2M**	**9**	**2**	**512**	**512**
IS-Net	3	16	0.9917	0.9891	48K	9	2	512	512
	3	8	0.9826	0.9828	0.12M	9	2	512	512
	3	32	0.9933	0.9936	1.9M	9	2	512	512

**Table 3 jimaging-08-00139-t003:** A comparison between the original U-Net and the compressed U-Net (proposed). The prediction time (in seconds) is observed on CPU only for a single B-scan. Abbreviations: VDSC—validation DSC, TDSC—test DSC, MS—model size, TT—training time, PT—prediction time. The entries corresponding to the proposed network are highlighted in bold.

Network Arch.	Stage	# Parameters	VDSC	TDSC	MS(Mb)	TT	PT
U-Net [22]	RS	31M	0.9986	0.9985	121	114	1.97
	IS	31M	0.9935	0.9929	121	129	2.06
ResU-Net [46]	RS	13M	0.9986	0.9985	52	112	1.08
	IS	13M	0.9943	0.9944	52	130	1.51
Compressed	RS	**79K**	0.9979	0.9976	**0.4**	**18**	**0.11**
U-Net	IS	**1.2M**	0.9933	0.9933	**5**	**51**	**0.31**

**Table 4 jimaging-08-00139-t004:** Training-related metrics. Abbreviations: NetA—network architecture Tr—training, Te—test, Val—validation, Acc—accuracy.

NetA	Tr Acc	Tr DSC	Tr Loss	Val Acc	Val DSC	Val Loss	Te Acc	Te DSC	Te Loss	Epoch
RS	0.9985	0.9992	0.0064	0.9984	0.9992	0.0064	0.9985	0.9991	0.0061	28
IS	0.9953	0.9974	0.0175	0.9952	0.9974	0.0179	0.9950	0.9973	0.0185	11

**Table 5 jimaging-08-00139-t005:** A comparison between HeyEx segmentation and the proposed method (CCU-INSEG) using MAE (standard deviation) in μm. Abbreviations: HeyEx—Heidelberg eye explorer (device integrated segmentation software).

Method	ILM	RNFL-GCL	IPL-INL	INL-OPL	OPL-ONL	ELM	EZ-OS	BM	Total
HeyEx	3.5 (3.2)	6.3 (2.6)	6.6 (2.5)	7.6 (2.5)	4.8 (3.0)	3.5 (2.8)	5.2 (2.9)	4.0 (2.7)	5.2 (2.6)
CCU	2.1 (0.4)	3.0 (0.6)	2.6 (0.4)	2.8 (0.5)	2.6 (0.7)	1.7 (0.4)	1.7 (0.4)	1.9 (0.6)	2.3 (0.4)

**Table 6 jimaging-08-00139-t006:** DSC between CCU-INSEG’s outcome and the manually segmented data.

Data	Vitreous	mRNFL	GCIPL	INL	OPL	ONL	MZ	OL	b-BM	Total
CUB	0.99	0.92	0.95	0.92	0.89	0.96	0.90	0.97	0.99	0.95
UCLA	0.99	0.89	0.94	0.92	0.88	0.96	0.88	0.97	0.99	0.94
JHU	0.99	0.90	0.94	0.84	0.83	0.94	0.85	0.95	0.99	0.92

**Table 7 jimaging-08-00139-t007:** MAE (standard deviation) in μm between CCU-INSEG’s outcome and the manually corrected segmentation for each boundary.

Data	ILM	RNFL-GCL	IPL-INL	INL-OPL	OPL-ONL	ELM	EZ-OS	BM	Total
CUB	2.1 (0.4)	3.1 (0.6)	2.5 (0.4)	2.9 (0.4)	2.6 (0.7)	1.8 (0.4)	1.7 (0.4)	1.8 (0.4)	2.3 (0.4)
UCLA	2.2 (0.2)	4.2 (1.0)	2.7 (0.4)	3.2 (0.5)	2.9 (0.4)	2.0 (0.3)	2.0 (0.4)	2.0 (0.5)	2.6 (0.3)
JHU	3.2 (0.4)	4.0 (0.7)	3.9 (0.6)	5.3 (1.0)	3.8 (1.0)	2.6 (0.4)	2.6 (0.4)	4.0 (0.9)	3.7 (0.3)

**Table 8 jimaging-08-00139-t008:** The reliability of the standard parameters (top three rows) and fidelity of the computed parameters (MAE in μm) from CCU-INSEG’s segmentation with respect to the manual segmentation (the last three rows). Abbreviations (The last two letters are either T or V, where T is thickness and V is volume. The other abbreviations are as follows: ICC—intra-class correlation coefficient, LCI—lower confidence interval, UCI—upper confidence interval, mRNFL—macular retinal nerve fiber layer, GCIPL—macular ganglion cell and inner plexiform layer, INL—inner nuclear layer, GCC—ganglion cell complex, TM—total macula).

Param	mRNFL T	GCIPL T	INL T	GCC T	TM T	mRNFL V	GCIPL V	INL V	GCC V	TM V
ICC	0.93	0.99	0.96	0.99	0.99	0.93	0.99	0.96	0.99	0.99
LCI	0.88	0.98	0.93	0.97	0.98	0.88	0.98	0.93	0.97	0.98
UCI	0.96	0.99	0.98	0.99	0.99	0.96	0.99	0.98	0.99	0.99
CUB	1.2	0.9	0.8	0.5	0.7	0.024	0.018	0.016	0.010	0.014
UCLA	2.0	1.6	0.8	0.5	0.6	0.039	0.032	0.016	0.009	0.013
JHU	0.9	1.8	6.9	1.4	2.4	0.018	0.035	0.140	0.028	0.044

**Table 9 jimaging-08-00139-t009:** DSCs are computed using 45 manually delineated B-scan images between AURA, OCTExp, RelayNet and CCU. ★ RelayNet and OCTExp segment ONL + MZ in a single class; therefore, MZ column is empty for both, and ONL column shows DSC for both ONL and MZ. ▲ RelayNet segments OL in two different classes. For this comparison, we combined both classes in OL and computed DSC with respect to manual segmentation. The best outcomes for each layer are highlighted in bold.

Method	Vitreous	mRNFL	GCIPL	INL	OPL	ONL	MZ	OL	b-BM	Total
AURA [45]	**0.99**	0.57	0.62	0.86	0.85	**0.96**	**0.92**	**0.98**	**0.99**	0.86
OCTExp [49]	**0.99**	0.56	0.76	0.73	0.69	0.89 ^★^	-	0.85	**0.99**	0.80
RelayNet [20]	0.87	0.70	0.73	0.64	**0.86**	0.77 ^★^	-	0.90 ^▲^	0.92	0.80
CCU	**0.99**	**0.88**	**0.93**	**0.88**	0.84	**0.96**	0.89	**0.98**	**0.99**	**0.93**

**Table 10 jimaging-08-00139-t010:** A comparison between the proposed method (CCU, in bold) and the state-of-the-art methods in terms of the correction time (in seconds) and the deviation (in μm) between the corrected and uncorrected versions of the selected 15 volumes. MAE: mean absolute error. The best outcomes are highlighted in bold.

Method	Avg Correction Time (Min-Max) (s)	Correction Needed (Volumes)	MAE (Min–Max) (in μm)
HeyEx	162 (99–320)	11 (73.3%)	0.016 (0–0.301)
OCTExp [49]	209 (86–638)	6 (40%)	0.034 (0–0.950)
AURA [45]	596 (204–1658)	15 (100%)	0.164 (0–5.938)
CCU	**97 (82–150)**	**2 (13.3%)**	**0.001 (0–0.115)**

## Data Availability

The data presented in this study are available on request from the corresponding author. The data are not publicly available due to limitations from the European General Data Protection Regulations and German laws.

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
