# Peer review of "Intraretinal Layer Segmentation Using Cascaded Compressed U-Nets"

_2313-433X, 2022, doi:10.3390/jimaging8050139_

Round 1
Reviewer 1 Report
The paper is well presented and organized, making it easy to understand. The method is well motivated and demonstrates to be effective. The experiments are extensive. However, some minor issues should be further addressed:
- In Line 173, it presents that compressed u-net helps to balance processing needs and accuracy. However, what do the processing needs refer to? like processing speed? Though Table 3 presents some comparisons, there are no runtime comparisons between different versions of u-net.
- There is no ablation study to evaluate the performance of the model with only RS-Net or only IS-Net.
- Cascade reasoning is a popular idea, and the authors should make an more inclusive review of relevant papers such as Cascaded parsing of human-object interaction recognition.
Author Response
Please see the attachment, which includes point-by-point responses for both reviewer comments.

Reviewer 2 Report
The authors proposed a dep learning based method for intraretinal layer segmentation. Following are my comments:
1) I am not satisfied with the current form of the Abstract. The authors should describe the clinical problem first, then how existing methods are lacking, and then present their approach.
2) Introduction is satisfactory but it can be enhanced by adding more recent references
3) What are the main differences b/w Casecade compressed U-Net and ordinary U-net?
4) I would suggest to add a comparison table between U-Net variants and Proposed method
5) Add the reference for pre-and post-processing schemes
6) change pixel-imbalance to class-imbalance?
7) I would suggest a statistical performance test to enlight the performance difference (example T-test) for Table 3 and where possible
Author Response

(The authors gave the same response as above.)

Round 2
Reviewer 1 Report
The revision has addressed most of my concerns. Thus I recommend it for acceptance.
Reviewer 2 Report
The authors responded to the comments correctly, therefore I vote for acceptance of this paper in its current form.